# Self-Assessment Instruments for Supporting Family Caregivers: An Integrative Review

**DOI:** 10.3390/healthcare12101016

**Published:** 2024-05-14

**Authors:** Franzisca Domeisen Benedetti, Mareike Hechinger, André Fringer

**Affiliations:** School of Health Professions, Institute of Nursing, ZHAW—Zurich University of Applied Sciences, Katharina-Sulzer-Platz 9, 8401 Winterthur, Switzerlandandre.fringer@zhaw.ch (A.F.)

**Keywords:** family caregivers, informal care, self-assessment, burden of care, caregiver burden, integrative review

## Abstract

Family caregivers take on a variety of tasks when caring for relatives in need of care. Depending on the situation and the intensity of care, they may experience multidimensional burdens, such as physical, psychological, social, or financial stress. The aim of the present study was to identify and appraise self-assessment instruments (SAIs) that capture the dimensions of family caregivers’ burdens and that support family caregivers in easily identifying their caregiving role, activities, burden, and needs. We performed an integrative review with a broad-based strategy. A literature search was conducted on PubMed, Google Scholar, Google, and mobile app stores in March 2020. After screening the records based on the eligibility criteria, we appraised the tools we found for their usefulness for family care and nursing practice. From a total of 2654 hits, 45 suitable SAIs from 274 records were identified and analyzed in this way. Finally, nine SAIs were identified and analyzed in detail based on further criteria such as their psychometric properties, advantages, and disadvantages. They are presented in multi-page vignettes with additional information for healthcare professionals. These SAIs have proven useful in assessing the dimensions of caregiver burden and can be recommended for application in family care and nursing practice.

## 1. Background

Taking on the important role of an informal caregiver is associated with considerable personal demands and a great social impact on the family caregivers of older people and, in the future, on national health and care systems [1]. Against the background of the challenges of a rapidly aging population in Switzerland and internationally, supporting and reducing the burden of family caregivers is of great importance [2,3].

Across the world, the number of older people (60+) is expected to have more than tripled by 2100, increasing from 901 million people in 2015 to 2.1 billion in 2050 and 3.2 billion in 2100 [4]. This means that the number of people in need of care will also increase. Informal care is a necessity for the care of the elderly in most countries and is even a cornerstone of long-term care *systems* in the region designated by the United Nations Economic Commission for Europe (UNECE), comprising 56 countries. Societies rely to varying degrees on the unpaid work of informal caregivers [5,6]. At the same time, the pool of family caregivers is likely to decrease, as the share of the working-age population in countries that are members of the Organization for Economic Co-operation and Development (OECD) is expected to shrink from 67% in 2010 to 58% by 2050. The average contribution of family caregivers varies significantly between countries depending on the definition and measurement modality [7]. In the EU, for example, family caregivers, mainly women, provide over 80% of all care [8].

We define family caregivers as persons who care for, look after, and, if necessary, assist a close relative who is ill and/or in need of care with personal hygiene and activities of daily living [9]. We use the term *family* to refer to closely related individuals and informal caregivers, regardless of family relationship [10]. Role theory provides additional guidance, explaining that family caregivers behave in different, predictable ways depending on their respective social identities (spouse, child, etc.) and the situation [11]. Self-assessment instruments (SAIs) may assess such aspects of caregiving. Deeken et al. have conducted a comprehensive literature review of SAIs to identify and critically appraise such instruments developed for research purposes [1]. This is in line with the policy agenda in Europe, which is to build knowledge bases in order to systematically generate measures supporting informal care [12]. 

Self-assessment, defined here as the process of exploring and evaluating oneself and aspects of caregiving, is important because it can enable self-awareness of the impact of family caregiving tasks. The tasks that family caregivers perform can be perceived as stressful and burdensome but can also have positive effects, such as personal growth. From a family system perspective, informal caregiving enables a deeper connection between the person being cared for, the family caregiver, and the broader family system, as well as the development of resilience and intimacy. On the other hand, caregiving has an impact on all that are involved, for example by disrupting home life and juggling competing roles [13]. The caregiver’s role is multifaceted and may be financial, administrative, and/or coordinative. Family caregivers also provide help with daily activities and household chores, as well as emotional and social support [9,14]. The degree of burden varies depending on the situation. Findings suggest that the amount of time spent and the intensity of care are key indicators of the experience of burden caused by caregiving tasks [9] (p. 43ff).

In addition to these indicators, the clinical picture of the relative in need of care is among the so-called objective stressors, which form the basis of stress models [15,16]. In Pearlin’s stress model [16,17], which guides our work, family caregivers’ stress can be viewed as the result of a process involving several interrelated conditions, including the socio-economic characteristics and resources of family caregivers and the primary and secondary stressors to which they are exposed. Primary stressors are hardships and problems that are directly rooted in caregiving. Secondary stressors are (1) the stresses experienced in roles and activities independent of caregiving activities and (2) intrapsychic stresses that negatively affect the self-concept. Consequently, the subjective experience of stress may differ for the same objective stress parameters (e.g., socio-economic characteristics), as the subjective evaluation of a stressful situation also depends on individual secondary stressors. 

Various dimensions of family caregivers’ burden that influence each other or are mutually dependent have been identified [9,14,18]. According to Otto et al. [9], these include, for example, being under time pressure and having little time and energy for themselves, psychological burdens such as stress, depression, excessive demands, or burnout, physical burdens such as pain from heavy lifting, social burdens due to family and/or role conflicts, loneliness, and social isolation, or financial burdens. Bastawrous [19] discusses the concept of “caregiver burden” and concludes that the multiple definitions of “caregiver burden” lead to vague findings that are difficult to summarize and appraise in a consistent manner. At the same time, the applicability of these findings in clinical and policy settings is limited. We followed Bastawrous’ [19] recommendation to use stress theory and role theory as guiding frameworks to capture the contextual features of caregiver burden that are relevant to caregiving outcomes. In doing so, we used Pearlin’s stress model to guide our work [16,17]. Moreover, by focusing on SAIs to highlight the unique experiences of each informal caregiver, we used role theory to facilitate our understanding of how “caregiver burden” can arise as a result of role conflict and role overload [11]. Thus, we defined “caregiver burden” broadly to capture its multidimensional aspects, such as burden, strain, insufficient coping, stress, and insufficient caregiver mastery. 

Coping strategies and social support by healthcare professionals (HCPs) can potentially address several points in the stress process. Therefore, understanding the nature of burden, measuring caregivers’ strain, and integrating this information into family care practice is crucial to effectively support family caregivers. To implement a family system focus, we need to use a systemic approach that aims at sustainably improving the everyday care management of families and enabling them to live their daily lives together in the home setting [20,21]. The aim is, therefore, to prevent family members from becoming overburdened and running the risk of falling ill themselves. To achieve this, a partnership is sought between the person being cared for, the family caregivers, and professionals [22]. 

In order to be able to offer, develop, and evaluate adequate services and support for family caregivers, it is essential to appropriately assess the situation of family caregivers in the home environment as a first step. SAIs can be used for this purpose. That is, SAIs support family caregivers in easily identifying their caregiving role, activities, burden, and needs. Therefore, the following research questions guided this study: (1) Which SAIs capture the dimensions of family caregivers’ burden? (2) Which SAIs can be recommended for designated use by family caregivers and to inform nursing support in family practice? Accordingly, the aims of this study are to identify SAIs, examine their purpose and key characteristics, and appraise SAIs that can successfully be applied in family care and nursing practice. We also aimed at developing instrument vignettes in order to generate guidance for HCPs. 

## 2. Methods

### 2.1. Study Design 

To address the first research question, we conducted an integrative review to find suitable instruments. We based our approach on Whittemore and Knafl’s [23] integrative review, which aims to include empirical and theoretical literature to create a comprehensive understanding of the topic [24].

An integrative review offers a unique advantage for identifying and evaluating SAIs for family caregivers by integrating and synthesizing both studies with different methodologies (experimental, non-experimental, qualitative, quantitative) and instruments used in family and nursing practice, providing a more comprehensive understanding of the effectiveness of tools in real-world caregiving scenarios. This is consistent with the emphasis on integrative reviews in nursing science due to their highlighting multiple perspectives [25]. Addressing methodological complexity is crucial. While methods of data collection and extraction have been developed, robust methods of analysis and synthesis specific to integrative reviews are still being developed [26]. This is particularly important when dealing with the large and diverse data sets, which are common in integrative reviews. Our study aims to contribute to this ongoing development by critically discussing the methods used.

To answer the second research question, we appraised the instruments found in the references using predefined evaluation criteria, to identify instruments that can successfully and usefully be applied in family care and nursing practice.

### 2.2. Search Strategy

This integrative review followed a broad strategy and was divided into three strings: (1) a systematic search in the PubMed database, (2) a hand search in Google Scholar, Google, reference screening in literature reviews, in German-language journals, and in different app stores, and (3) contacting selected institutions, organizations, and experts in the field of supporting burdened family caregivers to identify additional relevant published and unpublished research.

In strings 1 and 2, we used a multi-phase search process to first identify potentially relevant instruments in the literature. In string 3, we received information on the instruments used. We then searched for published reports on measures that had been assessed for their suitability. 

This comprehensive search strategy without time limits aimed to find instruments published in English, German, French, and Italian. The literature search took place in March 2020. The work of Deeken et al. [1] and the FOPH [2] served as the basis for the development and selection of the search terms, which were combined into search strings. The detailed search strategy can be found in Appendix A. The following key terms were used and adapted to the different databases: patient reported outcome measure OR self-report OR questionnaire AND needs assessment OR stress OR burden OR exhaustion AND caregiver OR family AND community health service OR home care service. We used MeSH terms and PubMed as the only database to streamline the search process, and we accepted that we might miss relevant studies using alternative terminology. We compensated for this with the hand search.

The systematic search in PubMed (1) was performed with English-language search terms. The hand search (2) was conducted in Google Scholar (English, German) and Google (German, French, Italian). Since Google search engines cannot be searched systematically, different combinations of search terms were used and the first 100 hits of each search run were screened [27]. Both the systematic search and the hand search were conducted independently by four people. To identify instruments that are used in family care and nursing practice, (3) selected organizations, institutions, and experts were also contacted via email. The people contacted were asked to report the self-assessment tools they used.

### 2.3. Record and Instrument Screening

The literature selection and data extraction were carried out in a three-step process, based on the method described by Kleibel and Mayer [28]: (1) title and abstract screening against the inclusion and exclusion criteria in Table 1, (2) review of full texts against the eligibility criteria, and (3) review of instruments in the remaining full texts against the eligibility criteria and screening the instruments. The inclusion and exclusion criteria are listed in Table 1. No restrictions of a methodological nature (e.g., only psychometrically tested instruments) were placed. All types of SAIs were included to allow for the inclusion of self-developed instruments and applications in app stores in addition to scientifically developed instruments [1]. SAIs developed exclusively for use in nursing homes were excluded, as the focus is on the home care situation of family caregivers. However, SAIs developed and used in the hospital setting were included as they can also be used by family caregivers in the home setting. SAIs were excluded if any of the content item exclusion criteria were met.

In the selection process, we use the term record because the individual hits in the search process include studies, grey literature, instruments per se, and apps. A flowchart of the literature search is shown in Figure 1 and illustrates the following three steps of the screening process, based on Kleibel and Mayer [28]. First, (1) records from the systematic search in PubMed were imported into a reference management system and independently screened by title and abstract by two people. Records that did not meet the inclusion and exclusion criteria were excluded. Afterwards, the records found in the systematic search and potentially matching records from the hand search were combined and duplicates were removed. In the second step, (2) the remaining records were imported into Excel and the corresponding full texts (if applicable) were retrieved. We extracted the following information for each record: author(s), title, year of publication, name of the instrument, abbreviation (if available). If a record included multiple instruments, multiple rows were created. The reasons for exclusion were noted in one column. Thirdly, (3) the instruments themselves were extracted, unless the instrument was already exclusively available, and checked again separately against the inclusion and exclusion criteria by two researchers. Discrepancies were discussed with a third researcher. For this step, we listed the instruments alphabetically and erased duplicates. We extracted, again in Excel, the following information for each instrument: name of the instrument, abbreviation (if available), developer, continent, target group, aim/purpose, and content elements. The reasons for exclusion were noted in one column. During this step, it became clear that there are few web-based instruments and not all of them have a score. It was therefore decided to discard the predefined inclusion criteria of instrument type in order not to exclude a priori any potentially important instruments that could potentially be converted into web-based forms in the future or that have the potential to stimulate reflection on the situation of family caregivers. Whittemore and Knafl [23] identify the following steps in their discussion of data analysis: data reduction, data display, and data comparison. In terms of the three steps described above, steps 1 and 2 were data reduction, data display comprised steps 2 and 3, while data comparison occurred in step 3.

### 2.4. Appraisal for Applicability in Family Care and Nursing Practice 

After the extraction of SAIs according to the inclusion criteria, an evaluation against the predefined content elements was conducted to identify and appraise useful instruments that can successfully be applied in family caregiving and nursing practice. This evaluation followed guideline No. 6 for “family caregivers of adults” [29] from the German Society of General Practice and Family Medicine (DEGAM). The DEGAM develops scientifically sound and practice-proven guidelines according to the principles of evidence-based medicine.

SAIs were assessed using evaluation criteria that included the following content components, based on the work of Pearlin et al. and Otto et al. [9,16]: (1) referral to activities/tasks as a family caregiver; (2) intensity of caregiving activities; (3) caregiver burden and positive impact of the caregiving situation; (4) caregiver support needs; (5) caregiver health status; (6) psychometric properties such as the use of a rating scale, summative outcome, and/or cut-off score. 

Since the focus was on SAIs, the studies or records from which the instruments were extracted were not subjected to a quality appraisal. Accordingly, no assessment was made of the descriptions of the SAIs there. Instead, the quality appraisal of SAIs was based on the content evaluation criteria. We selected SAIs which included content components (1), (2), and (3), were able to assess primary and secondary stressors, and (6) allowed for the quantification of self-assessed caregiving tasks as well as of the subjective burden and positive effects of caregiving. If SAIs met all 4 criteria, they were deemed very suitable for application in family caregiving and nursing practice. 

After appraising and selecting SAIs suitable for family caregiving and nursing practice in that way, we analyzed them in depth against the following criteria: psychometric properties such as number of items, rating scale/sum score, cut-off score, and validation of the respective SAI; conclusion on the applicability of the SAI in family care and nursing practice; as well as advantages and disadvantages. To generate guidance for HCPs, instrument vignettes were developed for instruments that were analyzed in depth. To develop the instrument vignettes, we consulted additional literature in order to be able to give a comprehensive picture of the respective instruments. Each multi-page vignette comprises a part giving an overview of the SAI and a part providing more in-depth information (see Appendix A).

## 3. Results

A total of 2654 publications were identified based on a systematic literature search of the PubMed database, a hand search, and from the contacts made with institutions, organizations, and experts. The systematic search in PubMed yielded n = 2139 potential instruments via publications, the hand search n = 5, and contacting institutions, organizations, and experts yielded n = 10. A total of 681 full texts were reviewed against the eligibility criteria. From the remainder of n = 495 publications, n = 1004 instruments were identified. After removing duplicates and screening the instruments for eligibility, n = 45 instruments from a total of n = 274 records could finally be included. The flowchart of the literature search is shown in Figure 1.

### 3.1. Results: Instruments for Assessing Burden in Family Caregivers 

Our integrative and systematic search resulted in 45 instruments that fulfilled the eligibility criteria. The following information relates to the contents listed in Table 2, which provides an overview of the characteristics of each included instrument. The instruments included are from 274 records listed in Appendix A.

To answer our first research question, we identified 45 instruments that capture the dimensions of the family caregiver burden associated with caregiving tasks and their intensity. These instruments were mainly developed in North America (n = 26) and in Europe (n = 16), followed by Asia (n = 5) and Australia (n = 1).

Regarding the target group, the majority of the instruments focus on adult family caregivers regardless of their age (n = 32) and one instrument each addresses caregiving parents of sick children and caregiving spouses. Another seven instruments refer to caregivers without any age specification and only four instruments are explicitly suitable for older caregivers aged 50 and older. Instruments specifically target family caregivers of older adults (n = 10), individuals with neurological conditions including dementia or Alzheimer’s disease (n = 14), cancer (n = 5), psychiatric (n = 4), palliative (n = 3) cardiac (n = 1), or hematologic conditions (n = 1). Only one instrument targeted family caregivers of children (n = 1). Eight instruments are without specification.

The instruments include items about the negative (such as burden, strain, stress) and/or positive (such as personal growth) effects of family caregiving. All forty-five instruments address negative effects, and three instruments (Nr. 16: Caregiver Tasks Inventory; Nr 25: Carers of Older People in Europe—COPE Index; Nr. 31 Hemophilia Caregiver Impact measure) address both. They also focus on different dimensions of burden. Some assess objective burdens such as caregiving tasks or intensity of care, and others focus on subjective burdens such as stress and other conditions resulting from caregiving.

Forty-one instruments had a score. Because we included instruments regardless of their psychometric properties, it is important to keep in mind that the presence of a score is not equated with good psychometric properties. Thirty instruments were related to the tasks that family caregivers perform. At least 15 instruments referred to the health status of the family caregiver and 15 instruments included the intensity of care or support effort; only 13 referred to needs or support needs. Some of the instruments included were developed for professional or scientific use.

Among the instruments included is the Carers Assessment of Difficulties Index (CADI). This instrument was not found to be useful for practice. However, in combination with the Carers Assessment of Satisfactions Index (CASI) and the Caregiver Assessment Management Index (CAMI), it would meet the inclusion criteria as a combination of tools that is comprehensive but may help to clarify complexity. Because we were looking for single tools, we did not include the CASI-CADI-CAMI combination. Similarly, both the Multidimensional Assessment of Caring Activities (MACA) and the Positive and Negative Outcomes of Caring (PANOC) [77] were excluded as each of them did not fulfill the eligibility criteria. MACA focuses on tasks but is not related to burdens, while PANOC focuses on caring outcomes. If one follows the recommendation to use MACA and PANOC combined, burdens and positive outcomes are considered. Still, these instruments are intended for professional or scientific use. We were looking for instruments that are easy to understand and suitable as SAIs. The MACA and PANOC are easy to understand but must be used with guidance from a professional.

### 3.2. Results: Applicability in Family Care and Nursing Practice

To answer the second research question, the included n = 45 SAIs were first analyzed in terms of their content components. Nine SAIs were judged to be particularly suitable for use in family caregiving practice as they included the content components we selected as our evaluation criteria: (1) caregiving tasks, (2) intensity of caregiving activities, (3) caregiver burden and/or positive effects of the caregiving situation, and (6) psychometric properties such as sum score, rating scale, or cut-off score. Table 3 describes the output of the in-depth analyses of these nine SAIs. They are listed in alphabetical order according to instrument name (abbreviation in parentheses); the authors and the respective publications are mentioned.

As seen in Table 2, all included instruments (n = 45) focus on different dimensions of burden. Some assess objective burdens, such as the time spent on caregiving tasks and their intensity; others focus on subjective burdens such as psychological stress and other conditions resulting from caregiving.

The following SAIs can be used to measure mainly psychological stress (≥ 50% of the items): the Caregiver Self Assessment Questionnaire—CSAQ; the Burden Scale for Family Caregivers—BSFC; the Caregiving Health Engagement Scale—CHE-s; the Family Caregiver Distress Assessment Tool; and the Caregiving Appraisal Scale—CAS. The following SAIs can be used to measure psychological stress (< 50% of the items) as well as physical, socio-economic, and/or temporal stress: the Zarit Burden Interview—ZBI; the Caregiver Strain Index—CSI; and the Caregiver Burden Inventory—CBI. The psychometric properties, such as number of items, rating scale/sum score, and cut-off score, of the respective SAIs are shown in Table 3. As far as validation is concerned, seven out of the nine SAIs have been validated, mainly in English. The Zarit Burden Interview (ZBI) has been most frequently translated, culturally adapted, and validated in several languages.

To sum up, these instruments (n = 5) are short and easy-to-understand questionnaires that can be used independently by family caregivers. At the same time, they all have a high level of acceptance. All instruments have the potential to depict family caregiving as a multidimensional process that takes into account the time and the developmental, physical, social, and emotional issues of family caregivers.

In conclusion, aspects of the applicability of these SAIs in family care and nursing practice as well as their advantages and disadvantages are shortly described in Table 3; detailed instrument vignettes can be found in Appendix A. No digital web-based application, which would further facilitate assessment and outcomes through automation, is available among the instruments (n = 7). All instruments more or less explicitly recommend that their application be supported by nursing or social work professionals (or other healthcare professionals) to interpret the self-assessment results and plan appropriate support measures.

## 4. Discussion

The aims of the present integrative review were to identify SAIs that capture content elements such as the burden of family caregivers, appraise the suitable instruments, develop instrument vignettes in order to generate guidance for HCPs, and verify the applicability of SAIs for family caregivers and nursing practice.

Our practical contribution is an evidence-based, systematic approach to searching, screening, and evaluating the identified SAIs for their use in family caregiving and nursing practice. We have presented each of the nine final SAIs in a multi-page vignette that provides professionals with an immediate, condensed overview of each SAI (Appendix A). Our theoretical contribution is a discussion of the integrative review method.

Kirkevold [25] argued that integrative nursing research can improve the development of nursing science and make research-based knowledge more accessible to clinical nurses. In particular, integrative reviews have become increasingly popular in the field of nursing. Hopia et al. [82] saw the potential of integrative reviews for the nursing field, but at the same time emphasized a systematic approach.

For Elsbach and van Knippenberg [83], the benefit of the integrative review is that it goes beyond a mere summary of the literature and adds something new through critical analysis. Integrative reviews can both consolidate evidence and generate new ideas to advance a field of study. They further argue that insights should arise from the integrative review rather than guide it. In this context, they advocate a more open approach. This can of course be seen as a contradiction to the required systematic approach. Alvesson and Sandberg [84] critique conventional views of integrative reviews and propose an alternative approach called a problematizing review, which emphasizes reflexivity, selective reading, problematizing the existing literature, and the concept of “less is more”. The problematizing review is seen as an “opening exercise” that generates new and better ways of thinking about particular phenomena or issues. Given this, we will discuss the practical contribution made by this review.

After a sensitive literature search, a total of 45 SAIs were identified. Nine contained all the content elements we deemed important for practical application to family care and nursing practice. The authors of these instruments describe the assessment of the extent of burden as the main purpose of the respective self-assessments, and it is assumed that all nine instruments are suitable for analyzing the extent of perceived burden. The demand to develop a corresponding digital, web-based instrument is strengthened by the results.

Thus, SAIs can help in the systematic identification of strengths and socio-economic, physical, and psychological risks in order to suggest tailored support and to initiate a personality development process. The results of qualitative studies indicate that the positive aspects of informal caregiving are also related to personal growth [85,86]. This includes the feeling of competence and accomplishment of difficult tasks. The sense of accomplishment also relates to skills and relationships, for example, having a closer relationship with and being able to give back to the person being cared for as well as discovering inner strengths through connecting with others. Feeling gratitude can also be a positive aspect of caregiving [85,86]. HCPs can support caregivers in identifying the positive aspects of caring and developing their personal strengths.

Concerning negative aspects, caregivers identify physical and emotional stress as well as feeling unprepared or unsupported as central challenges [86]. In their systematic review, Bom et al. [87] conclude that the included studies indicate a causal negative impact of caregiving on physical and mental health. The subgroup of married female caregivers, in particular, appears to experience negative health effects of caregiving [87]. Consultations with HCPs play a critical role in collaborative reflection and in the selection of an appropriate intervention. For example, an individual family caregiver’s burden profile may be the outcome of the application of an SAI that can be discussed during a consultation with a HCP, as early as possible, and that can give the HCP a chance to verify whether and how their interventions have had an impact on the family caregiver’s situation [79].

For family caregivers, self-assessment has been found to be useful for self-reflection and self-awareness and provides a basis for discussion with professionals. A study in a palliative care unit reported very positive experiences [88]. According to the study, the SAI used provided direction, focus, and structure for discussion with professional caregivers and identified the needs of family caregivers.

The results of a meta-analysis by Sörensen et al. [89] demonstrated a potential for the improvement of caregiver burden. However, spousal caregivers benefited less than adult children. Individually tailored interventions have been shown to be more effective at improving caregiver well-being. A systematic review by Lopez-Hartmann et al. [90] shows that the effects of caregiver support interventions are small and inconsistent between studies. They propose interventions tailored to the caregivers’ individual needs. Technology-based interventions can have a positive effect on caregiver self-efficacy, self-esteem, and burden [91]. In a rapid review of systematic reviews, Spiers et al. [92] concluded that the current evidence fails to determine how caregivers should be supported and that further studies are needed to identify appropriate interventions. We suggest that HCPs should focus on vulnerable subgroups, such as married female caregivers, and use SAIs as tools to support the reflection process and identify individually tailored interventions. SAIs are also becoming increasingly important in scientific discourse and exist for different diseases and settings. From an interprofessional perspective, family caregivers, home care professionals, and general practitioners experience psychological burden as the most serious form of burden. General practitioners and nurses cited a lack of interprofessional education, lack of time, and lack of compensation as the main problems. Family caregivers valued communication with primary care physicians and nurses [93]. Empowering family caregivers to use SAIs may be one way to address this gap in care.

It should be noted that many of the instruments found that looked promising at first sight often had gaps, and thus may require a combination of instruments for the respective area of application. The methodological approach of the present study was deliberately integrative in order to identify instruments used both in research and in practice. Accordingly, app stores were also searched. None of the identified apps were analyzed in more detail, not because they were not user-friendly, but because essential content aspects were missing. In general, it should be noted that there are few web-based or digitally available tools. In a review, Firmawati et al. identified mobile applications for family caregivers of people with stroke as supportive [94]. Studies have found evidence that mHealth or eHealth tools for family caregivers have an impact on the burden of care [95].

Over the past 45 years, researchers have developed SAIs to assess family caregivers. Many studies have shown that applying SAIs has an impact on identifying different dimensions of burden [1]. This integrative review has identified valuable and promising instruments that, with translation, cultural adaptation, and web-based support, could be applied in a low-threshold way in order to identify caregiver burdens early and accompany the challenges of care with the support of HCPs. It can be also stated that there is a lack of empirical studies investigating the relationship between regular self-assessment of family caregivers and burden reduction.

### Strengths and Limitations

In light of the methodological discussion above, we have opted for a systematic approach. Nevertheless, our formulation of purpose was not quite as specific as Hopia et al. [82] recommend. In addition, we used the integrative review method to find SAIs. This meant that the target of our search were both the instruments themselves and studies through which these instruments were obtained. With this decision, we expanded the classic approach. In order to achieve all our objectives, we divided our approach into several sub-steps, presented our actions in detail, and provided further information on our review in Appendix A. Due to our broad strategy, we only included Google Scholar and PubMed as scientific databases. More scientific databases could have been included. In the spirit of reflexivity, we acknowledge that free terms in addition to MeSh terms would have been useful. As the literature search was conducted in 2020, possible new or updated SAIs may not have been taken into account. Since minor changes to the inclusion and exclusion criteria in the context of reviews can mean serious differences in results [24], these should be critically considered by each reader and questioned for their own context of application. One of the strengths of this integrative review is our four-language search strategy. Our broad approach, as well as our constant discussion process in the preparation of the present study, can be seen as strengths. Since literature studies are only as good as the data, specifically the publications included, publication bias cannot be ruled out [24].

## 5. Conclusions and Implications

In principle, the present work on identifying and evaluating self-assessment instruments can be regarded as an important first step to strengthen theoretical and applied knowledge. The integrative method was used to identify instruments used for research and practice. As for the application of these instruments in practice, the question arises as to how the most appropriate instruments can be made available and accessible to the family caregivers.

SAIs have the potential to increase family caregivers’ self-awareness of their role, the resulting burden, and the need for professional support. There are indications that low-threshold dissemination to family caregivers via care providers or information services is possible [42]. In order to clarify the when, how, and why, it is important to understand the interactions between the national structural order, the institutional forces, their impact on people’s well-being, and people’s physical and mental health and their ability to assert and develop themselves in their social roles (e.g., as family caregivers). To promote understanding and accompany the introduction of self-assessment instruments into practice, it is recommended to consult basic concepts such as stress process models [15,16], dynamic models [96], or caregiver identity theories [97]. Not only general practitioners or family doctors but also community health nurses are in a suitable position to apply the best-fitting instrument. One of the core competences of the community health nurse, for example in Germany, but also in other countries, is the application of assessments in outpatient care [98].

It is obvious that further studies and research efforts are needed in order to be able to determine the specific cut-off values and the resulting effective interventions. It is also central to find out what the uses and acceptance levels of the individual SAIs will be for family caregivers in different settings, on the one hand, and in communities on the other. Especially for professionals who are in contact with family caregivers, multidimensional instruments, such as assessments of subjective and objective stress dimensions, are of particular interest. Our integrative review covers the period up to 2020; an update would be useful for the period after that. Furthermore, the development of digital and web-based instruments is an important implication for settings, countries, or institutions that want to support family caregivers. Furthermore, through this review, the SAIs identified in Appendix A can be a basis for further intervention development research.

## Figures and Tables

**Figure 1 healthcare-12-01016-f001:**
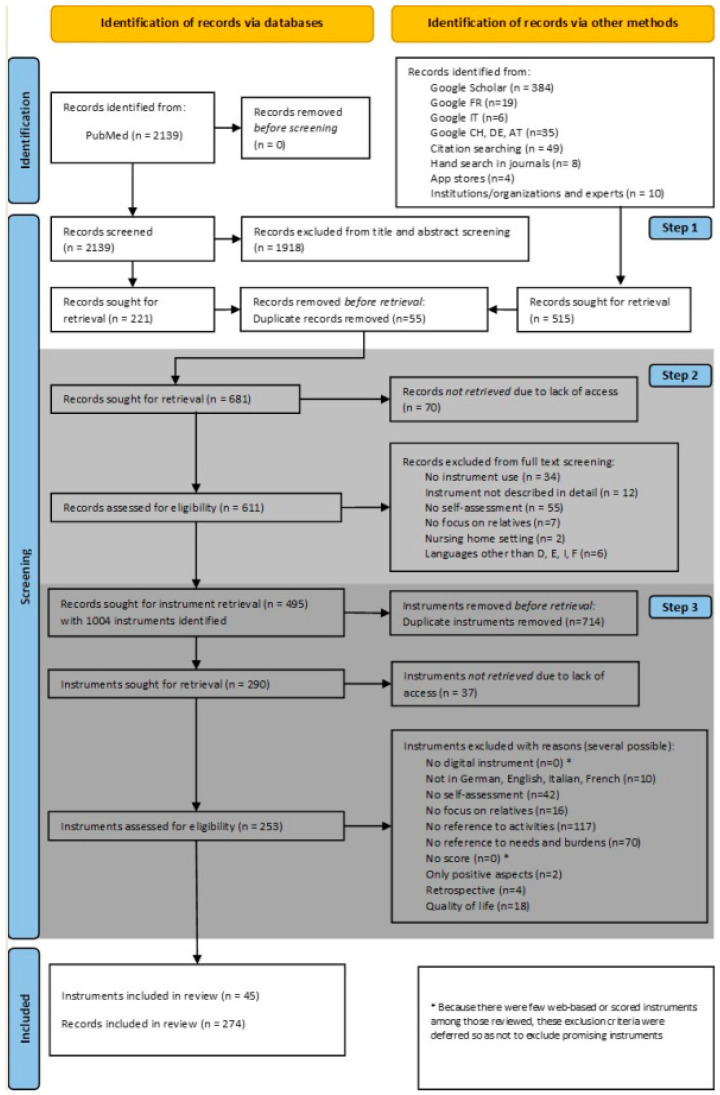
Flowchart of the literature search.

**Table 1 healthcare-12-01016-t001:** Inclusion and exclusion criteria for full text analysis and eligible instruments.

	Inclusion Criteria	Exclusion Criteria
Type *	- instrument that is digitalized/web-based - instrument that has a score, so that recommendations for the relatives can be derived	- instrument that is not digitalized/web-based - instrument that has no score so that no recommendations can be derived
User Individuals	- family caregivers and/or - family, relatives, friends	- volunteers - exclusively HCPs or other professionals (e.g., social workers)
Setting	- home care situation and/or - hospital	- nursing home
Suitability	self-assessment, self-check, self-report, or self-recording instrument that - can be used independently and - is easy to understand	the instrument - consists only of open questions - can only be used with the advice of professionals or - can only be used together with professionals.
Content elements	- one or more activities of family caregiving and/or - burden and positive effects of caregiving and/or - family caregiver needs assessment and/or - relation to the group of persons in need of care (e.g., elderly persons) and/or - less than 25% illness-specific questions about the person being cared for	- only collects information needs - only assesses family needs that do not relate to care - focuses on caregiving satisfaction or quality of life - only records the time spent (quantitatively) on the care and support of relatives, e.g., as a diary function without a focus on burdens - includes disease-specific questions, >25% of which are related to the person being cared for

* Because there were few web-based or scored instruments among those reviewed, these exclusion criteria were deferred so as not to exclude promising instruments.

**Table 2 healthcare-12-01016-t002:** Overview of the included instruments (n = 45).

Nr.	Instrument	Abb.	Authors	Continent	Target Group	Aim/Purpose	Content Elements
**1**	German assessment to survey resources of and risks to elderly family caregivers	ARR	Budnick, A.; Kummer, K.; Blüher, S.; Dräger, D. [30]	Europe	Family caregivers > 50 years of age who are claiming benefits for home care through long-term care insurance for the first time.	The ARR is an instrument to capture the resources of and risks to older family caregivers with derivation of health promotion needs and corresponding activity suggestions.dervi	√ Care tasks (indirect) ☐ Intensity of care/support effort √ Burdens/positive effects of care √ Needs ☐ Support needs √ Health status √ Score
**2**	Brief Assessment Scale for Caregivers	BASC	Glajchen, M.; Kornblith, A.; Homel, P.; Fraidin, L.; Mauskop, A. [31]	Asia/North America	Adult caregivers (family members, friends, related persons) of persons with chronic illnesses (cancer, neurological, psychiatric).	The BASC is a brief instrument that can help identify caregivers with high burden.	☐ Care tasks ☐ Intensity of care/support effort √ Burdens/positive effects of care ☐ Needs ☐ Support needs ☐ Health status √ Score
**3**	Brief Assessment Scale for Caregivers		Pearlin, L.I.; Mullan, J.T.; Semple, S.J.; Skaff, M.M. [16]	North America	Adult caregivers (family members, friends, related persons).	The goal of this instrument is to provide a sound measurement of the many aspects of caregiving carried out by family members and their effects. For this purpose, a stress process model was developed to capture the numerous components of the phenomenon with various measurements.	√ Care tasks (√ Intensity of care/support effort) √ Burdens/positive effects of care ☐ Needs ☐ Support needs √ Health status √ Score
**4**	Burden Assessment Scale	BAS	Reinhard, S.C.; Gubman, G.D.; Horwitz, A.V.; Minsky, S. [32]	North America	Adult caregivers (family members, friends, related persons) of persons with a serious mental illness.	The BAS measures the objective and subjective burden of family caregivers who have to learn to cope with the symptoms of illness of the cared-for person. The scale distinguishes between caregivers with different levels of burden and is sensitive to changes over time.	√ Care tasks ☐ Intensity of care/support effort √ Burdens/positive effects of care ☐ Needs ☐ Support needs ☐ Health status √ Score
**5**	Burden Assessment Schedule	BAS	Thara, R.; Padmavati, R.; Kumar, S.; Srinivasan, L. [33]	Asia	Families of mentally ill people.	The BAS aims to measure both objective and subjective burden on family caregivers.	☐ Care tasks ☐ Intensity of care/support effort √ Burdens/positive effects of care ☐ Needs ☐ Support needs ☐ Health status ☐ Score
**6**	Burden Scale for Family Caregivers	BSFC	Grässel, E.; Leutbecher, M. [34]	Europe	Adult caregivers who care for someone at home with a chronic need for assistance or care (personal care, etc.). The care and support activity needs to have already existed for half a year.	The BSFC can be used to obtain a quick overview of the stress situation of family caregivers and aims to find and establish suitable and, above all, accepted relief measures (e.g., improving family support) for family caregivers at home. The measurement of stress can be carried out once or several times during the course of caregiving.	√ Care tasks √ Intensity of care/support effort √ Burdens/positive effects of care ☐ Needs ☐ Support needs √ Health status √ Score
**7**	Cancer Caregiving Tasks, Consequences and Needs Questionnaire	CaTCoN	Lund, L.; Ross, L.; Groenvold, M. [35]	Europe	Adult caregivers (family members, friends, related persons) of people with cancer.	The CaT-CoN questionnaire captures caregiving tasks, their consequences, and the needs of family caregivers, focusing on interactions (information sharing, communication, and contact) with health professionals.	√ Care tasks ☐ Intensity of care/support effort √ Burdens/positive effects of care √ Needs √ Support needs ☐ Health status ☐ Score
**8**	Caregiver Burden Inventory	CBI	Novak, M.; Guest, C. [36]	North America	Adult caregivers (family members, friends, related persons) of cognitively impaired (elderly, 65+) persons. The authors recommend that the CBI be transferred to different settings and also be used with family caregivers of persons with different chronic illnesses.	The CBI is an instrument that measures the impact of burden on family caregivers.	√ Care tasks √ Intensity of care/support effort √ Burdens/positive effects of care ☐ Needs ☐ Support needs ☐ Health status √ Score
**9**	Caregiver Burden Scale	CBS	Elmståhl, S.; Malmberg, B.; Annerstedt, L. [37]	Europe	Adult caregivers (family members, friends, related persons) of elderly stroke patients three years after a primary stroke or with dementia.	The CBS measures the burden on family caregivers.	☐ Care tasks √ Intensity of care/support effort √ Burdens/positive effects of care ☐ Needs ☐ Support needs ☐ Health status √ Score
**10**	Caregiver Burden Screen	CBS	Rankin, E.D.; Haut, M.W.; Keefover, R.W.; Franzen, M.D. [38]	North America	Adult caregivers (family members, friends, related persons) of persons with dementia.	The CBS establishes clinically relevant cut-off points for existing instruments that measure the burden on family caregivers.	☐ Care tasks ☐ Intensity of care/support effort √ Burdens/positive effects of care ☐ Needs ☐ Support needs √ Health status √ Score and cut-off for the risk calculation of care arrangements
**11**	Caregiver Needs Screen	CNS	Boele, F.W.; Terhorst, L.; Prince, J.; Donovan, H.S.; Weimer, J.; Sherwood, P.R.; Lieberman, F.S.; Drappatz, J. [39]	Europe/North America	Family caregivers of individuals with a malignant brain tumor with neurological and cognitive symptoms.	The CNS focuses on the needs family caregivers may face when caring for a loved one with a brain tumor.	☐ Care tasks ☐ Intensity of care/support effort √ Burdens/positive effects of care √ Needs ☐ Support needs ☐ Health status √ Score
**12**	Caregiver Reaction Assessment	CRA	Given, C.W.; Given, B.; Stommel, M.; Collins, C.; King, S.; Franklin, S. [40]	North America	Adult caregivers (family members, friends, related persons) of elders with physical disabilities, Alzheimer’s disease, and cancer.	The CRA is a multidimensional instrument for assessing family caregivers’ risk of excessive burden.	☐ Care tasks √ Intensity of care/support effort √ Burdens/positive effects of care ☐ Needs ☐ Support needs ☐ Health status √ Score
**13**	Caregiver Risk Screen	CRS	Guberman, N.; Keefe, J.; Fancey, P.; Nahmiash, D.; Barylak, L. [41]	North America	Adult caregivers (family members, friends, related persons) of people receiving professional home care.	The CRS proposes a systematic assessment of the situation of family caregivers. The purpose of this screening instrument is to assess in which area of physical and/or psychological well-being the family caregivers are at risk and whether the care provided is appropriate. A level or threshold of risk is determined to establish the urgency of intervention. The CRS shows promise for practical use in assessing changes in the situation.	√ Care tasks ☐ Intensity of care/support effort √ Burdens/positive effects of care ☐ Needs ☐ Support needs √ Health status √ Score
**14**	Caregiver Self Assessment Questionnaire	CSAQ	American Medical Association, AMA [42]	North America	Adult caregivers (family members, friends, related persons) of elderly persons.	The questionnaire can help family caregivers assess their own behavior and health risks (burdens).	√ Care tasks √ Intensity of care/support effort √ Burdens/positive effects of care ☐ Needs √ Support needs √ Health status √ Score
**15**	Caregiver Strain Index	CSI	Robinson, B.C. [43]	North America	Adult caregivers (family members, friends, related persons).	The purpose is to identify stresses in specific at-risk populations (with stressful caregiving relationships, emotional distress) of family members at an early stage.	√ Care tasks √ Intensity of care/support effort √ Burdens/positive effects of care ☐ Needs ☐ Support needs √ Health status √ Score
**16**	Caregiver Tasks Inventory		Clark, N.M.; Rakowski, W. [44]	North America	Adult caregivers (family members, friends, related persons).	The Caregiver Tasks Inventory was created based on caregiver tasks reported by family caregivers and a categorization of these tasks was created. (Positive) effects of education and support programs for family caregivers are also analyzed. This is used to characterize the burden of caregiving tasks.	√ Care tasks ☐ Intensity of care/support effort √ Burdens/positive effects of care √ Needs √ Support needs ☐ Health status √ Score
**17**	Caregiver’s Burden Scale in End-of-Life Care	CBS-EOLC	Dumont, S.; Fillion, L.; Gagnon, P.; Bernier, N. [45]	North America	Adult caregivers (family members, friends, related persons) of persons in the palliative phase.	The CBS-EOLC is an instrument that specifically assesses family caregiver burden in the context of palliative care.	☐ Care tasks ☐ Intensity of care/support effort √ Burdens/positive effects of care ☐ Needs ☐ Support needs √ Health status (√ Score)
**18**	Caregiving Appraisal Scale	CAS	Lawton, M.P.; Kleban, M.H.; Moss, M.; Rovine, M.; Glicksman, A. [46,47]	North America	Family caregivers of older persons with any type of impairment, dementia, and those at the end of life.	The CAS focuses on subjective evaluations of care: subjective burden, satisfaction with care, and perceived effects of care.	√ Care tasks √ Intensity of care/support effort √ Burdens/positive effects of care ☐ Needs ☐ Support needs ☐ Health status √ Score
**19**	Caregiving Hassles Scale	CHS	Kinney, J.M.; Stephens, M.A.P. [48]	North America	Adult, primary caregivers of individuals diagnosed with Alzheimer’s dementia and living at home.	The CHS focuses on minor everyday events, the mundane experience of caring and being cared for, and the minor irritations of daily life. Such stresses can be both temporary and long-lasting, and they are assessed by a caregiver as affecting their well-being.	√ Care tasks √ Intensity of care/support effort √ Burdens/positive effects of care ☐ Needs ☐ Support needs ☐ Health status √ Score
**20**	Caregiving Health Engagement Scale	CHE-s	Barello, S.; Castiglioni, C.; Bonanomi, A.; Graffigna, G. [49]	Europe	Family caregivers of individuals with complex care needs.	The CHE-s was developed to assess the psychosocial experience of caregiving and family caregivers’ engagement. The CHE-s shows the extent to which a balance is achieved between their caregiving tasks and their overall life goals. The CHE-s is intended to close the gap between what family caregivers experience in their everyday caregiving and what provides them with the most support.	√ Care tasks √ Intensity of care/support effort √ Burdens/positive effects of care ☐ Needs ☐ Support needs ☐ Health status √ Score
**21**	Caregiving Stress Appraisal Scale	CSA	Abe, K. [50]	Asia	Adult caregivers (family members) of persons receiving insurance benefits for long-term care.	The CAS scale is a simple instrument for measuring caregiver burden.	☐ Care tasks ☐ Intensity of care/support effort √ Burdens/positive effects of care ☐ Needs ☐ Support needs √ Health status √ Score
**22**	Carer Experience Scale	CES	Al-Janabi, H.; Coast, J.; Flynn, T.N. [51,52]	Europe	Adult caregivers (family members, friends, related persons) from diverse cultural backgrounds and cared-for individuals with diverse medical conditions.	The CES indicates which statement best describes the current caregiving situation.	√ Care tasks ☐ Intensity of care/support effort √ Burdens/positive effects of care ☐ Needs √ Support needs ☐ Health status √ Score
**23**	Carers Assessment of Difficulties Index	CADI	Nolan, M.R.; Grant, G. [53]	Europe	Adult caregivers (family members, friends, related persons).	The CADI helps professionals provide appropriate support strategies and services for family caregivers that address their needs.	√ Care tasks (indirect) ☐ Intensity of care/support effort √ Burdens/positive effects of care √ Needs ☐ Support needs ☐ Health status √ Score
**24**	Carer’s Checklist		Hodgson, C.; Higginson, I.; Jefferys, P. [54]	Europe	Adult caregivers (family members, friends, related persons) of people with dementia who receive specialized, professional care and by volunteers.	The Carer’s Checklist for Family Caregivers is used to elicit the extent of dementia-related problems in daily life and the burden they create, to assess the needs and changes in needs of people with dementia and their family caregivers, and to evaluate the outcomes of measures of care.	√ Care tasks √ Intensity of care/support effort √ Burdens/positive effects of care ☐ Needs ☐ Support needs ☐ Health status √ Score
**25**	Cope-Index	COPE	McKee, K.J.; Philp, I.; Lamura, G.; Prouskas, C.; Oberg, B.; Krevers, B.; Spazzafumo, L.; Bień, B.; Parker, C.; Nolan, M.R.; Szczerbinska, K. [55]	Europe	Adult caregivers (family members, friends, related persons) of elderly persons.	The COPE Index is an instrument for a brief initial assessment of family caregivers with questions about negative and positive outcomes, quality of care, and financial issues.	√ Care tasks (indirekt) ☐ Intensity of care/support effort √ Burdens/positive effects of care ☐ Needs ☐ Support needs √ Health status ☐ Score
**26**	Coping Inventory	CI	Barusch, A.S. [56]	Asia	Adult caregivers (family members, friends, related persons) of elderly persons with dementia.	The Coping Inventory includes dimensions with negative changes in the relationship between the cared-for person and the caregiver relative, limitations in the caregiver relative’s social activity, negative changes in family relationships, psychological distress, financial and economic distress, and health distress.	√ Care tasks √ Intensity of care/support effort √ Burdens/positive effects of care ☐ Needs ☐ Support needs ☐ Health status (√ Score)
**27**	Family Appraisal of Caregiving Questionnaire for Palliative Care	FACQ-PC	Cooper, B.; Kinsella, G.J.; Picton, C. [57]	Australia	Adult caregivers (family members, friends, related persons) of individuals in home palliative care.	The FACQ-PC is a multidimensional questionnaire to assess the impact of caregiving/caregiving by family members of persons in a palliative situation (FACQ-PC).	√ Care tasks ☐ Intensity of care/support effort √ Burdens/positive effects of care ☐ Needs ☐ Support needs √ Health status √ Score
**28**	Family Burden Scale	FBS	Madianos, M.; Economou, M.; Dafni, O.; Koukia, E.; Palli, A.; Rogakou, E. [58]	Europe	Adult caregivers (family members) of individuals with schizophrenia spectrum disorder.	The FBS records objective and subjective stresses of family caregivers who live at home with persons suffering from schizophrenia and are cared for by professionals.	☐ Care tasks ☐ Intensity of care/support effort √ Burdens/positive effects of care ☐ Needs ☐ Support needs √ Health status √ Score
**29**	Family Caregiver Distress Assessment Tool		Home Instead, [59]	North America	All family caregivers caring for an elderly person.	The Family Caregiver Distress Assessment Tool is used to record and assess the stress of family caregivers. It helps to find out which aspects might make caregiving more difficult and what can be done to meet these challenges.	√ Care tasks √ Intensity of care/support effort √ Burdens/positive effects of care ☐ Needs ☐ Support needs ☐ Health status √ Score
**30**	Heart Failure Caregiver Questionnaire	HF-CQ	Strömberg, A.; Bonner, N.; Grant, L.; Bennett, B.; Chung, M.L.; Jaarsma, T.; Luttik, M.L.; Lewis, E.F.; Calado, F.; Deschaseaux, C. [60]	Europe/North America	Adult caregivers (family members, friends, related persons) of older persons with heart failure.	Family caregivers of people with severe heart failure often suffer from great stress. The HF-CQ measures subjective aspects of burden for family caregivers of persons with heart failure.	√ Care tasks ☐ Intensity of care/support effort √ Burdens/positive effects of care ☐ Needs ☐ Support needs √ Health status √ Score
**31**	Hemophilia Caregiver Impact measure	HCI	Schwartz, C.E.; Powell, V.E.; Eldar-Lissai, A. [61]	North America	Adult caregivers (family members, friends, related persons) of persons with hemophilia A or B.	The HCI assesses the negative and positive effects of caregiving tasks on family caregivers of persons with hemophilia A or B.	√ Care tasks ☐ Intensity of care/support effort √ Burdens/positive effects of care ☐ Needs ☐ Support needs √ Health status √ Score
**32**	Impact of Event Scale (-Revised)	IES-R	Weiss, D.S.; Marmar, C.R. [62]	North America	Elderly adult caregivers of persons with long-term care needs.	The IES-R assesses psychological symptoms, such as post-traumatic stress disorder (PTB), related to a specific traumatic event (e.g., cancer diagnosis or treatment).	☐ Care tasks ☐ Intensity of care/support effort √ Burdens/positive effects of care ☐ Needs √ Support needs ☐ Health status √ Score
**33**	Impact on Family Scale	IOFS	Stein, R.E.K.; Riessman, C.K. [63]	North America	Parents of children with a chronic illness.	The IOFS can be used to study the impact of the care of chronically ill children on their families.	√ Care tasks (indirect) ☐ Intensity of care/support effort √ Burdens/positive effects of care ☐ Needs ☐ Support needs ☐ Health status √ Score
**34**	Life Situation among Spouses after a Stroke Event Questionnaire	LISS-Q	Larson, J.; Franzén-Dahlin, Å.; Billing, E.; Murray, V.; Wredling, R. [64]	Europe	Spouses of persons with a stroke living in the same household.	The LISS-Q is an instrument designed to assess the living situation of spouses of individuals who have had a stroke.	√ Care tasks (indirect) ☐ Intensity of care/support effort √ Burdens/positive effects of care ☐ Needs ☐ Support needs ☐ Health status √ Score
**35**	Modified Caregiver Strain Index	MCSI	Thornton, M.; Travis, S.S. [65]	North America	Older adult caregivers (family members, friends, related persons) of individuals with long-term care needs.	The Modified Caregiver Strain Index (MCSI) assesses caregiver strain following the discharge of an elderly family member from the hospital.	√ Care tasks ☐ Intensity of care/support effort √ Burdens/positive effects of care ☐ Needs ☐ Support needs √ Health status √ Score
**36**	Montgomery Borgatta Caregiver Burden scale	MBCBS	Montgomery, R.J.; Borgotta, E.F. Montgomery, R.J.; Gonyea, J.G.; Hooyman, N.R. [66,67]	North America	Adult caregivers (family members, friends, related persons) of individuals with long-term care needs.	The Montgomery Borgatta Caregiver Burden scale emphasizes the importance of distinguishing between objective and subjective burdens on family caregivers.	√ Care tasks ☐ Intensity of care/support effort √ Burdens/positive effects of care √ Needs ☐ Support needs ☐ Health status √ Score
**37**	Needs Assessment of Family Caregivers-Cancer	NAFC-C	Kim, Y.; Kashy, D.A.; Spillers, R.L.; Evans, T.V. [68]	North America	Adult caregivers (family members, friends, related persons) of individuals who are cancer survivors.	The NAFC-C helps identify the needs of family caregivers of cancer survivors to improve their quality of life, which can be affected not only during diagnosis but long after treatment.	√ Care tasks ☐ Intensity of care/support effort √ Burdens/positive effects of care √ Needs ☐ Support needs ☐ Health status √ Score
**38**	Parent Caregiver Strain Questionnaire	PCSQ	England, M.; Roberts, B.L. [69]	North America	Adult children of a parent with a neurological impairment.	The PCSQ measures the level of caregiver task exhaustion, emotional distress, and goal deviation distress.	√ Care tasks ☐ Intensity of care/Support effort √ Burdens/positive effects of care ☐ Needs ☐ Support needs ☐ Health status √ Score
**39**	Perceived Stress Scale	PSS	Cohen, S.; Kamarck, T.; Mermelstein, R. [70]	North America	Diverse groups of family caregivers.	The PSS is the most widely used psychological instrument for measuring stress perception. It is a measure of the extent to which situations in one’s life are perceived to be stressful. The questions were developed to capture how unpredictable, uncontrollable, and overloaded family caregivers perceive their lives to be.	☐ Care tasks ☐ Intensity of care/support effort √ Burdens/positive effects of care ☐ Needs ☐ Support needs ☐ Health status √ Score
**40**	Sekentei scale for caregivers	SSC	Asahara, K.; Momose, Y.; Murashima, S.; Okubo, N.; Magilvy, J.K. [71]	Asia	Adult caregivers (family members, friends, related persons) of elderly persons in need of care.	The SSC was developed to describe the phenomenon of “Sekentei” for family caregivers in Japan. “Sekentei” is a social psychological process that restricts behaviors in the family that do not conform to social norms such as family caregiving. In addition, the relationships between “Sekentei” and caregiving relatives who actually claim services or refuse to claim services and the impact/burden were described.	☐ Care tasks ☐ Intensity of care/support effort √ Burdens/positive effects of care ☐ Needs ☐ Support needs ☐ Health status √ Score
**41**	Self-test for family caregivers		Seniorplace [72]	Europe	Adult caregivers (family members, friends, related persons).	Seniorplace offers an online self-assessment instrument to assess the own health of family caregivers. Based on the respective information, the burden is visualized with the help of a traffic light system.	☐ Care tasks ☐ Intensity of care/support effort √ Burdens/positive effects of care ☐ Needs √ Support needs ☐ Health status √ Score: not obvious which instrument
**42**	Self-developed unmet need measure		Gaugler, J.E.; Anderson, K.A.; Leach, M.S.W.C.R.; Smith, C.D.; Schmitt, F.A.; Mendiondo, M. [73]	North America	Elderly caregivers (family members, friends, related persons) of persons with dementia in home care and also after the cared-for person transfers to a nursing home or dies.	The starting point is that the emotional burden of caregiving relatives persists after the termination of home care (e.g., nursing home admission). This instrument captures various dimensions of unmet needs that have an influence on subjective stress.	☐ Care tasks √ Intensity of care/support effort √ Burdens/positive effects of care √ Needs ☐ Support needs ☐ Health status (√ Score)
**43**	Sense of competence questionnaire Short Sense of competence questionnaire	SCQ SSCQ	Vernooij-Dassen, M.J.; Felling, A.J.; Brummelkamp, E.; Dauzenberg, M.G.; van den Bos, G.A.; Grol, R. [74]	Europe	Adult caregivers (family members, friends, related persons) of persons with dementia.	The SCQ and SSCQ support the identification of stress in family caregivers as a first step.	√ Care tasks ☐ Intensity of care/support effort √ Burdens/positive effects of care ☐ Needs ☐ Support needs ☐ Health status √ Score
**44**	Social Support Rating Scale	SSRS	Xiao, S. [75]	Asia	Family caregivers	Social support is a factor that can reduce stress. This is measured in the SSRS. Higher scores indicate better social support.	☐ Care tasks ☐ Intensity of care/support effort √ Burdens/positive effects of care ☐ Needs √ Support needs ☐ Health status ☐ Score
**45**	Zarit Burden Interview	ZBI	Zarit, S.H.; Reever, K.E.; Bach-Peterson, J. [76]	North America	Adult primary caregivers (family members, friends, related persons) of older persons with dementia and other conditions who are cared for at home.	The ZBI incorporates behaviors of the cared-for person, the relationship of the caregiver to the cared-for person, and support from other family members that contribute to the burden of home care and caregiving.	√ Care tasks √ Intensity of care/support effort √ Burdens/positive effects of care ☐ Needs ☐ Support needs ☐ Health status √ Score

**Table 3 healthcare-12-01016-t003:** Most suitable SAIs (n = 9) for application in family care and nursing practice.

Instrument	Author(s)	Psychometric Properties	+ Advantage − Disadvantage	Conclusion for Applicability
**Burden Scale for Family Caregivers (BSFC)**	Grässel, E.; Leutbecher, M. [34]	Number of items: long version: 28; modified short version: 10. The statements to be assessed in the questionnaire refer to the type of assistance that caring relatives give. There are statements to be assessed in connection with care tasks and their intensity: e.g., item 28, “In addition to support/alongside care, I can carry out my other tasks of daily life according to my expectations”. These statements can apply to support, care, or nursing. A total score can be calculated. The questions are rated on a scale from 0 = do not apply to 3 = apply. The total score ranges from 0 to 84 points. Higher scores indicate a greater burden on family caregivers. Validated in German and English.	+ easy to understand and can be completed by family caregivers themselves in 5–10 min + available in a short version (BFSC-s) with ten questions + calculates a sum value and is a valid measure of the total burden on family caregivers + suitable for use in practice and in research − no web-based version available yet	The use of the BSFC/HPS helps users to reflect on what their family support network looks like, what can be improved, and what support services by HCPs can bring relief to their situation; The BSFC/HPS is suitable for independent use by family caregivers. The questionnaire is available as a paper version with a template for calculating a sum score and thus facilitates the calculation of the total burden; Application to parents of sick children and children with disabilities is possible; A digitized version would enable the automatic calculation of the total burden. Potential digitization and the possibilities of use must be clarified with the authors.
**Caregiver Burden Inventory (CBI)**	Novak, M.; Guest, C. [36]	Number of items: 24. Each question receives a score between 0 = not applicable and 4 = very applicable, with higher scores indicating a greater burden on the relatives; there are no cut-off values for the classification of burden. Therefore, total scores for dimensions one, two, four, and five can range from 0 to 20. An equivalent score for physical burden is obtained by multiplying the sum of the responses by 1.25. Validated in English and Italian.	+ includes short, simple questions that can be answered independently by family caregivers in 10–15 min + quantifies stresses in different areas of caregivers’ lives + enables the creation and interpretation of an individual Caregiver Burden Profile (CBP) + multidimensional and takes into account time and developmental, physical, social, and emotional issues of family caregivers − no web-based version available yet	Family caregivers may complete the CBI questionnaire on their own. The developers recommend using the CBI with the help of professionals (e.g., social counseling, initial assessment in home care); The derivation of appropriate support measures requires a high level of expertise and consulting skills from professionals in nursing or social work [78,79]; The individual stress profile (evaluation and assessment of the result) can then be discussed during a consultation, as early as possible [78,79]; It is advisable to apply the CBI early on, e.g., when family caregivers first assume their role and are informed about offers of external support. The CBI can then be repeated over the course of caregiving. This also makes it possible to verify whether and in what way the measures taken have had an impact on the situation of the family.
**Caregiver Self Assessment Questionnaire (CSAQ)**	American Medical Association, AMA [42]: Online as pdf available	Number of items: 18. Family caregivers are asked to assess the situation in the past week with the following: − 16 statements about the level of effort of caregiving tasks (“I could not leave my relative alone”), psychological stress and positive effects of caregiving, and their health status. These can be answered with either “yes” or “no.”; − 1 question on the current degree of stress on a scale of 1 to 10, with 1 = not stressful and 10 = extremely stressful; − 1 question on the current state of health compared to the situation one year ago on a scale of 1 to 10, with 1 = very healthy and 10 = very sick. A total score can be calculated to identify health risks (e.g., indications of exhaustion and depression). In the online version, the total score is calculated automatically, and family caregivers receive their result and pre-score directly, as well proposals on how to proceed. No scientific validation, but appraised as a valid self-assessment instrument for detecting depressive symptoms in family caregivers [80].	+ easy-to-use online questionnaire available + family caregivers are encouraged to take a moment to reflect on their own well-being + can be completed at any point during the course of a relative’s care + before anyone completes the online questionnaire, they are advised to be sure to contact a healthcare professional − written permission from the Health in Aging Foundation must be obtained for use; use might be withheld in other areas of the world	The questionnaire can help family caregivers to assess their own behavior and the resulting psychological, physical, time, and social burdens. In addition, there is a question on stress level and a question on health-related well-being; The short online questionnaire is suitable for independent use by family caregivers. Following the recommendations of the American Medical Association (AMA), the questionnaire can be used as a guide. The results of the questionnaire should be discussed with a physician; The result can be discussed not only with a doctor, but also with nursing and social work professionals; Appendix A includes the results of a fictitious family caregiver who completed the online questionnaire; The online questionnaire is in English and refers to the American care situation. In addition, there is an Italian translation of the questionnaire as a paper pencil (PDF) version.
**Caregiver Strain Index (CSI)**	Robinson, B.C. [43]	Number of items: 13. The presence of objective (e.g., “sleep is disturbed (e.g., because … is in and out of bed or wanders around at night) ) and subjective distress can be answered with “yes” or “no”. Each “yes” receives one point and thus the maximum score is 13. The CSI has a cut-off point: a total score of 7 or more indicates a high degree of distress. Validated in English.	+ short, easy-to-use instrument (13 questions) to identify stress (e.g., shortly after taking up caregiving tasks) + calculates an overall score to detect high stress − no web-based version available yet	CSI and the Modified Caregiver Strain Index (MCSI) are self-assessment instruments that are easy and quick to implement and use and provide an indication of whether family caregivers are experiencing high levels of burden; It is commonly used in clinical practice; Both the CSI/MCSI can be used as a one-time screening tool to identify stresses in certain at-risk populations at an early stage. This includes, for example, individuals with stressful caregiving relationships or other emotional stresses; Application to parents of sick children and children with disabilities is possible; It is recommended that the CSI/MCSI be used in consultation with a caregiver or social work professional to assess the relationship between the family caregiver and the cared-for person; Permission of the publisher is required for use.
**Caregiving Appraisal Scale (CAS)**	Lawton, M.P.; Kleban, M.H.; Moss, M.; Rovine, M.; Glicksman, A. [46,47]	Number of items: 27, revised version available. 9 questions about the subjective burden of caregiving; 6 questions about satisfaction with caregiving; 6 questions about coping with caregiving; 3 questions about care/care requirements; 3 questions about impact on caregiving. Question type A asks the following question: “I would like to talk about some feelings you may have about caregiving for your mother, etc. Please tell me if you” with responses ranging from 1 = don’t agree at all to 5 = agree completely. Question type B asks the following question, “Tell me how often you feel each way”, with response categories 1 = never to 5 = almost always. A score can be calculated for each question area. This means that, in each area, answers are assessed as follows: burden of caregiving (high score means burdened); satisfaction with caregiving (high score means satisfied); coping with caregiving (high score means good coping); caregiving requirements (high score means demanding); impact on caregiving (high score means unfavorable impact). Validated in English.	+ contains a subjective assessment of all aspects of care (care tasks, care intensity/care effort) + examines the positive and negative effects of care + continuously developed; question categories and the scope of questions vary depending on the care situation and environment (setting) and the group of persons cared for. Therefore, a broad application is possible − no web-based version available yet	Exclusively used in studies and is being developed further. The developer also recommends its practical use. It can be included in surveys in different settings and to groups of family caregivers; Application to parents of sick children and children with disabilities is also possible; For application, support by nursing or social work professionals is needed to calculate and assess the result and to plan appropriate supportive measures; Derivation of suitable support measures requires a high level of expertise and consulting skills from the nursing or social work professional.
**Caregiving Hassles Scale (CHS)**	Kinney, J.M.; Stephens, M.A.P. [48]	Number of items: 42; in combination with the Uplift Scale: 110 questions. The questionnaire asks about events that have or have not occurred in the past week. If the event has occurred, then the respondents rate the extent of the stress on a 4-point scale from 1 = “not at all” stressful to 4 = “very stressful”. In combination with the Uplift Scale, the extent of the stress or the positive effect can be assessed on a 4-point scale ranging from 1 = “it was not (stressful/enriching)” to 4 = “very (stressful/enriching)”. Validated in English.	+ determines the general level of stress among family caregivers, e.g., from specific sources of stress − very extensive, with 42 questions or 110 questions when used with the Uplift Scale − no web-based version available yet	This is a diagnosis-specific instrument. It has so far been used for relatives of persons with dementia. From our point of view, it can also be used by caregivers of persons with other disabilities and parents of sick children and children with disabilities; It is useful to use the CHS in combination with the Uplift Scale to record the positive effects of the previous week’s care in addition to the stress; Independent completion of the questionnaire by family caregivers is demanding; The CHS/Uplift Scale provides a detailed picture of the day-to-day care experience. To use it, support from care or social work professionals is needed to calculate and assess the result and to plan appropriate supportive measures; Derivation of appropriate support measures requires a high degree of professional and advisory competence from the nursing or social work professionals.
**Caregiving Health Engagement Scale (CHE-s)**	Barello, S.; Castiglioni, C.; Bonanomi, A.; Graffigna, G. [49]	Number of items: 7. The family caregiver rates 7 items in the dimensions “management of personal time”, “psycho-physical stress”, “emotional and social concerns”. They can choose the statement that best describes their current experience from four statements. Each choice theoretically corresponds to one of the items described in the CHE model (1 = denial, 2 = overactivity, 3 = overload, 4 = balance). Lower scores correspond to the “denial” and “overactivity” positions of the CHE model, while higher scores correspond to the “overload” and “balance” positions. The results indicate how stress is shaped, e.g., how the family caregiver behaves and feels in the situation. More details about the scoring system are available from the authors upon request. Validated in Italian.	+ indicates that the engagement of family caregivers is a psychosocial process that results from a dynamic path of maturation and redefinition of the role of the individual family member in the course of accompanying and caring for the person being cared for + it is possible, on the one hand, to assess the level of commitment of family caregivers and, on the other hand, to better tailor supportive and educational measures to their needs − no web-based version available yet	The CHE-s provides a detailed picture of the day-to-day care experience and what measures provide relief. Family caregivers can answer CHE-s questions independently; Application requires good support from nursing or social work professionals to calculate and assess the outcome and plan appropriate supportive interventions; The derivation of suitable support measures requires a high level of expertise and consulting skills from the nursing or social work professionals; For use in Switzerland, the CHE must be translated into German and French, culturally adapted, and digitized.
**Carer’s Checklist**	Hodgson, C.; Higginson, I.; Jefferys, P. [54]	Number of items: 35. The Carer’s Checklist consists of two parts. The first part contains a list of 30 dementia-related problems that may occur. Each problem is asked about, and the following answers should be provided: − Frequency of occurrence of the problem: How often does the problem occur in the person being cared for? Answers: 0 = never; 1 = sometimes; 2 = always, with a point maximum of 60; − Subjective stress: How stressful do caregivers rate the problem? Answers: 0 = not stressful; 1 = quite stressful; 2 = very stressful, with a score maximum of 60. The second part of the checklist consists of five assessments (1 = not at all stressed to 5 = severely stressed) of overall stress, physical stress, financial stress, emotional stress, and social stress, with a maximum score of 25. Scores can be calculated individually for each part and repeated over time. Has not been validated in studies [81].	+ easy to use and takes about 15 min to complete independently + high level of acceptance of use among professionals and family caregivers − dementia-specific and should only be used in cases where there is confirmed diagnosis of dementia − no web-based version available yet	Family caregivers could answer the Carer’s Checklist questions independently. The questions are easy to answer; To assess the individual stress situation (taking into account information about the person being cared for) and to take appropriate measures, it is advisable to seek the advice of specialists; Repeated completion of the checklist can provide useful information about the development of the care situation as it allows for the comparison of results; Total point values can be calculated manually. With digitalization, it would be possible to calculate the result automatically. For practical use, a digital solution would be useful; Assessing the outcomes and planning appropriate supportive measures requires consultation with nursing or social work professionals and a high level of expertise and consultation skills.
**Zarit Burden Interview (ZBI)**	Zarit, S.H.; Reever, K.E.; Bach-Peterson, J. [76]	Number of items: long version: 22; short versions: 1, 4, 7, 12. After brief instructions, family caregivers are asked to answer a series of questions about how the illness of the person they care for has impacted their own lives. For each item, they indicate how often they feel this way (0 = never, 1 = sometimes, 2 = sometimes, 3 = quite often, or 4 = almost always). The ZBI is scored by adding up the numbered responses of the individual statements. Higher scores indicate a greater burden on family caregivers. Burden levels are estimates from a previous work. These are as follows: 0–20 little or no burden; 21–40 light-to-moderate burden; 41–60 moderate-to-heavy burden; 61–88 heavy burden Validated in English, German, Italian, French, and other languages.	+ supportive interventions for families and professional support networks have been described, e.g., “network meetings,” discussion groups such as self-help groups, scheduling of home visits by family members, friends, and neighbors + widely used in practice for different target groups and care contexts + available in many languages, including German, French and Italian + different validated short versions are available, with 1 to 12 questions − it is reported from practice that relatives are sometimes frightened by the extent of the burden − time required to complete is approximately 25 min	Family caregivers can answer the ZBI independently or as part of a conversation. In order to assess the individual stress situation (taking into account information about the cared-for person) and to take appropriate measures, it is advisable to seek the advice of professionals; It is an advantage if the professionals know the instrument well. Knowledge of the connection between frequent family member visits and the reduction in the burden of primary caregivers is of particular importance. In this way, measures can be taken to include other relatives in the close environment of the disabled person (other family members, friends, neighborhood) and their resources; It should not be used as the sole indicator of the caregiver’s emotional state. Clinical observations and other instruments, such as, e.g., the measurement of depression, should be used as a supplement; It has been shown that the ZBI successfully differentiates between different target groups and is suitable for measuring longitudinal developments. Application to parents of sick children and children with disabilities is possible; Since its development, the ZBI has been used in practice for a variety of target groups and care contexts and is available in many languages—including German, French and Italian.

## Data Availability

See Appendix A.

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
