# Peer review of "Self-Assessment Instruments for Supporting Family Caregivers: An Integrative Review"

_healthcare, 2024, doi:10.3390/healthcare12101016_

Round 1

Reviewer 1 Report

Comments and Suggestions for Authors

Respected Authors

The authors investigated the self-assessment instruments for supporting family caregivers through an integrative review. Family caregivers play an important role in families and studying this category of people in the service delivery system is of great importance. Therefore, the results of such studies are important because few studies have specifically addressed the actual needs of this population. 

- Abstract, please remove subheading from your abstract. Please also take a look at the journal guidelines in this regard. 

- Abstract, methods, please add more essential steps of methods in the abstract section. you only mentioned one step of search in this type of study. You can benefit from these five stages based on the literature. The 5-stage integrative review process includes (1) problem formulation, (2) data collection or literature search, (3) evaluation of data, (4) data analysis, and (5) interpretation and presentation of results.

-Abstract, please also add more details regarding your results. The way you report the whole abstract and results section is incomplete. You only state the results of the search here. Please add relative results based on the questions that introduce in the end of the background.

- Abstract, the conclusion is too general. Without conducting such a review, it is obvious that Various self-assessment instruments to assess dimensions of family caregiver burden are avail-26 able that can be recommended for application in family care and nursing practice. Please prepare a specific conclusion based on your aim and your results and remove this general statement.

- Keywords, please add "integrative review" as a keyword.

- Some parts of the manuscript have different fonts or sizes. Line 59, Reference 12.

- Line 72-99, big paragraph! Please break this paragraph into two paragraphs or summarize it in half. 

- Please re-write the final paragraph of the background section and end It with your aims. 

- Lines 1125-6, integrative review or systematic review? these two are different. 

- The steps of an integrative review are different of a systematic review. How do you use PRISMA when this checklist is not fit to your methodology? Also, if you use PRISMA for reporting this study why do you not follow it? The first item of PRISMA in the methods section is "Eligibility criteria" not "Search strategy".

- Line 135, wrong statement: "systematic and integrative review". Please refine it. 

- Line 136, "Three string" is correct. You need to benefit from a native editor or an editor who is familiar with this type of study. 

- Your search strategy is not sensitive enough. You use only MeSh terms and not free terms. Recent studies showed that using strategies with only MeSh terms is not a good strategy and wide search and not adding correct studies waste our time in screening. Also, your strategies for searching Google Scholar and Google are not practical. Google search is not suitable for practical tasks and is not recommended.

- Why do you contact selected institutions, organizations, and experts? What are selected institutions, organizations, and experts? and how you select them.

- Why did you search the databases by four people?

- Your eligibility criteria are not complete. The vice verse of inclusion criteria is not exclusion criteria. 

- Your flow diagram is not complete. If you follow the PRISMA flow diagram, please go to the PRISMA statement page and select the right flow diagram for reporting the results of the search based on the 2020 update.

- Your search is not to date. The last time of your search was 2020, which is 4 years ago. There is a need to update the search and naturally, the results have changed significantly. In addition, the COVID-19 pandemic will also have many effects on this search and published studies.

- Table 2, no need to column 1. (Nr). No need to mention the Aim/purpose in this table. This table only includes the summary characteristics of the included studies. It is better to start with a column with the first author's name and its reference, year, country of origin, study design, type of instrument, target group, and elements. Also, you need to summarize this table. Also, you need to summarize Table 3. This type of presenting the result is not good enough. It is better to synthesize the results and avoiding to writing results as present in the original articles. 

- The discussion is not good and comprehensive. As I stated before, after your search the world experienced the COVID-19 pandemic and unfortunately you missed this period that has a great effect on your results and your population. You need to update your search and your discussion. 

- Considering the number of devices and the number of final articles, the number of references is small.

Best regards,

Comments on the Quality of English Language

There are some punctuation and grammatical errors throughout the text. It is recommended to benefit from a native editor who is familiar with this type of review. 

Cheers

Author Response

Dear reviewer 1

please find our responses to your comments in the attached pdf.

best regards!

Reviewer 2 Report

Comments and Suggestions for Authors

Comments to the authors: 

This article aimed to identify and appraise self-assessment instruments (SAI) that capture the dimensions of burden of family caregivers and that support family caregivers to easily identify their caregiving role, activities, burden and needs.  Given three major reasons: 1) family caregivers take on a variety of tasks in the context of caregiving for relatives in need of care; 2) depending on the situation and the intensity of care, they may experience multidimensional burdens, such as physical, psychological, social or financial stress; 3) we know little about the comparative aspects of self-assessment instruments (SAI), this article has the potential to add fresh knowledge to healthcare, nursing and informal family caregivers literature. However, the article raises several key issues and major revisions should be made in the current version.

First, although an integrative review was advocated as important for evidence-based practice initiatives in nursing, its rationale and benefits weren't mentioned throughout the article as required. In this regard, authors should consider that integrative reviews are popular in nursing because they use multiple sources of data to investigate the complexity of nursing practice. An integrative review addresses the current state of the evidence, the quality of the available evidence, identifies gaps in the literature, and suggests future directions for research and practice. Since not all of these issues are addressed in this article, they can have a rethink on these points.

Second, concerning the methods - 1) broad database (45 suitable SAI from 274 records) but from 2020 - this should be noted in the limitations chapter; 2) the literature search was only conducted in PubMed, Google Scholar, Google and App Stores - this limitation should be carefully noted and explained.  Further, including these limitations will help readers to understand the results of the review objectively. 

Third, very weak and underdeveloped are the chapters on Discussion, Conclusions and Implications.  A real theoretical contribution is missing, along with an applied contribution. How innovative are the findings compared to existing studies in the literature?

To conclude: This is an article with high potential to contribute, so it is worth allowing the authors to improve it. However, it is important to understand that this is a very basic integrative review. Therefore, the authors need to respond to the above issues and strengthen the theoretical and applied contributions of their study. This is written in an unconvincing way. For example, the current findings should be compared with previous findings and by considering previous studies capture the different dimensions of SAI (burdens of family caregivers). In particular, this should be developed in the discussion chapter.
I hope that my comments will enable authors to improve their work and thus publish a good article.

Good luck!

Comments on the Quality of English Language

no

Author Response

Dear reviewer 2

please find our responses to your comments in the attached pdf.

best regards!

Reviewer 3 Report

Comments and Suggestions for Authors

The Integrative Review presented here is carefully crafted. I particularly appreciate the clarity of its content and the practicality of its results.

Background

I would like to request a detailed definition of the term "self-assessment" mentioned in the title of the manuscript. Although it is briefly discussed in the introduction, a more detailed explanation seems necessary. This is due to the fact that some of the instruments included were not originally designed for self-completion, as self-assessment implies. An explanation of this discrepancy would therefore be useful.

Results

Please provide more detailed information on why (based on what criteria) 1918 records were excluded.

Including standardized information on the purpose of each instrument reviewed would increase the usefulness of the manuscript. Some instruments were designed for specific purposes, such as screening, individual diagnosis, or population-level prevalence monitoring. This influenced the type and wording of each item and could determine its applicability in different contexts. In addition, clearly stating the selection criteria for different types of instruments (i.e., scales, indices, and checklists) would help to clarify the scope of the review.

Another addition would be to indicate who the intended administrators of each instrument are. I suspect that in many cases the appropriate administrators may not be the caregivers themselves.

For the instruments listed in Table 3, adding basic descriptive information about each instrument (e.g., range of the scale, its mean in the original study, standard deviation, and AVE - average variance extracted) along with internal consistency and psychometric properties (e.g., Cronbach's alpha, RMSEA, SRMR, GFI, TLI) would enhance the usefulness of the review.

It would also be beneficial to identify the specific dimensions of the instruments involved. As only selected components of whole constructs are often used, explicit identification and description of these dimensions would increase the usefulness of the review.

The exclusion of the CASI-CADI-CAMI scales deserves further explanation. The discussion in paragraphs 277-283 still leaves some ambiguity about this decision.

Discussion/conclusion and implications

Please add information about the intended audience for your review. By identifying both specific research needs and practical application needs (e.g., development of interventions), the review can more directly address the interests of its audience.

Highlighting the potential benefits for caregivers themselves can also provide valuable insight into the practical implications of your review.

The manuscript is a valuable contribution and with the suggested improvements its impact could be even greater. I look forward to seeing the improved version.

Author Response

Dear reviewer 3

please find our responses to your comments in the attached pdf.

best regards!

Round 2

Reviewer 2 Report

Comments and Suggestions for Authors

Accept in present form!